# The Optimal Indication for Testosterone Replacement Therapy in Late Onset Hypogonadism

**DOI:** 10.3390/jcm8020209

**Published:** 2019-02-07

**Authors:** Yu Seob Shin, Jong Kwan Park

**Affiliations:** 1Department of Urology, Chonbuk National University Medical School, Jeonju 54907, Korea; ball1210@hanmail.net; 2Research Institute of Clinical Medicine, Chonbuk National University, Jeonju 54907, Korea; 3Biomedical Research Institute, Chonbuk National University Hospital, Jeonju 54907, Korea; 4Clinical Trial Center of Medical Device, Chonbuk National University Hospital, Jeonju 54907, Korea

**Keywords:** indication, hypogonadism, testosterone

## Abstract

The use of testosterone replacement therapy (TRT) for late-onset hypogonadism (LOH) is increasing every year; however, the literature shows that many men are using testosterone (T) without a clear indication. Previous studies have estimated that up to 25% of men who receive TRT do not have their T tested prior to initiation of the therapy. Given the growing concern and need for proper TRT, clinicians need evidence-based information that informs them on the optimal indication for TRT in LOH patients. The diagnosis of LOH requires the presence of characteristic signs and symptoms, in combination with decreased serum total testosterone (TT). Based on the recent guidelines by the International Society for the Study of Aging Male (ISSAM), the European Association of Urology (EAU), the European Society of Endocrinology (ESE), the European Academy of Andrology (EAA), and the American Association of Urology (AUA), a TT of 250–350 ng/dL is the proper threshold value to define low T. The optimal indication for TRT in LOH is the presence of signs and symptoms of hypogonadism, and low T without contraindications for TRT.

## 1. Introduction

Late-onset hypogonadism (LOH) is characterized by low testosterone (T) levels and clinical symptoms [1,2,3,4,5,6]. Sexual symptoms and fatigue are the earliest and most common presentations [1,2,3]. Other symptoms include depression, sleep alterations, poor concentration, and metabolic disorders are seen at borderline T levels [4,5,6]. The Total testosterone (TT) and free testosterone (FT) concentrations decrease with increasing age in men [7]. LOH is the result of a gradual drop in T; a steady decline in T levels of about 1% per year is well documented in men [8]. However, decreases in T concentrations with age are gradual and vary between individuals, with higher rates of decline in men with adiposity and comorbid diseases [7]. Testosterone has anabolic effects that include growth of muscle mass and strength, increased bone density and strength, and stimulation of linear growth, production of blood, and bone maturation [9]. In LOH, the anabolic effect of T is reduced due to its deficiency.

Hypogonadism can be divided into two major categories [10]. Primary hypogonadism, also known as primary testicular failure, originates from a problem in the testicles. Secondary hypogonadism indicates a problem in the hypothalamus or the pituitary gland, parts of the brain that signal the testicles to produce testosterone. LOH has the combined features of both primary and secondary hypogonadism.

Demographic data clearly demonstrate that the percentage of older adults in the general population is increasing [11]. Androgen deficiency in the aging male has become a topic of increasing interest and debate throughout the world [12]. Furthermore, social interest in testosterone replacement therapy has increased over the past few decades. To provide appropriate guidelines for optimal indication of treatment for LOH in aging men, we summarized the recent guidelines by the International Society for the Study of Aging Male (ISSAM), the European Association of Urology (EAU), the European Society of Endocrinology (ESE), the European Academy of Andrology (EAA), the American Association of Urology (AUA), and the Canadian Men’s Health Foundation.

## 2. Diagnosis

### 2.1. ISSAM Guidelines

The diagnosis of hypogonadism requires the presence of characteristic signs and symptoms (evidence level: Level 2, Grade A), in combination with decreased serum concentration of testosterone [13]. Symptoms of hypogonadism may be categorized as sexual or non-sexual. Sexual symptoms include erectile dysfunction, diminished frequency of morning erections, and decrease in sexual thoughts (low libido), as well as difficulty in achieving orgasm and reduced intensity of orgasm (Table 1) [14]. Non-sexual symptoms include fatigue, impotence, impaired concentration, depression, and decreased sense of vitality and/or well-being. Signs of hypogonadism also include anemia, osteopenia and osteoporosis, abdominal obesity, and metabolic syndrome [15]. Principally, the clinician has to distinguish between forms of congenital hypogonadism, which can be congenital (e.g., Kallmann syndrome, Klinefelter syndrome) or acquired (e.g., anorchia due to trauma or orchiectomy, pituitary lesions/tumors, or LOH), requiring lifelong hormonal therapy, and forms of hypogonadism that might be reversible [13]. The latter, potentially reversible forms of hypogonadism are most often found to co-existence with metabolic disorders, such as obesity/type 2 diabetes mellitus (T2DM), inflammatory diseases (e.g., chronic obstructive pulmonary disease, chronic inflammatory bowel diseases), and anemia, or psychological problems, such as depression or stress.

Screening questionnaires on male symptomatic hypogonadism, although sensitive, have low specificity. Morley et al. compared the most commonly-used questionnaires in 148 men using bioavailable testosterone (BT) as the biochemical ‘‘gold standard’’ for the diagnosis of hypogonadism and found the sensitivity to be 97% for the Androgen Deficiency in the Aging Male questionnaire (ADAM), 83% for the Aging Male’s Symptoms scale (AMS), and 60% for the Massachusetts Male Aging Study questionnaire (MMAS). Specificity was 30% for the ADAM, 59% for the MMAS, and 39% for the AMS [16]. Despite having low specificity, the AMS and other male hypogonadism questionnaires may be useful to assess the presence and severity of symptoms as a prerequisite for initiating and monitoring clinical response to testosterone replacement therapy (TRT) [13].

Physical examination of patients with suspected hypogonadism should include an assessment of the amount and distribution of body hair (including beard growth and pubic hair), presence of acanthosis nigricans, associated with insulin resistance, presence, and degree of breast enlargement, size and consistency of the testes, abnormalities in the scrotum and size, appearance of the penis, and presence of subcutaneous plaque [13]. The prostate should be examined in older patients for size, consistency, symmetry, and presence of nodules or induration; it should be noted that the prostate may be enlarged in older men, despite a low testosterone level [17]. Weight, height, body mass index (BMI), and waist circumference should also be measured, since signs and symptoms potentially indicative of T deficiency in men include height loss, reduced muscle bulk and strength, and increased body fat, particularly increased BMI and abdominal fat accumulation [18].

In patients at risk or suspected of hypogonadism, a thorough physical and biochemical work-up is recommended (evidence level: Level 2, Grade A). The key laboratory tests to confirm the diagnosis of hypogonadism are serum TT and FT. Transient decreases of serum T levels can occur due to acute illnesses [19], and this should be excluded by careful clinical evaluation and repeated hormone measurements.

The ISSAM guidelines recommend 12.1 nmol/L as the lower limit of normal total testosterone [13]. However, due to individual differences in T sensitivity, some men may exhibit symptoms of hypogonadism with TT concentrations above this threshold and may benefit from TRT. TRT may be reasonably offered to symptomatic men with T concentrations lower than 12 nmol/L, based on clinical judgment. Although the ISSAM guidelines for FT levels have not yet been set, the guidelines mention that FT levels as low as 225 pmol/L (65 pg/mL) have been recommended as the normal range lower limit and, together with the presence of one or more hypogonadal symptoms, can provide supportive evidence for TRT (evidence level: Level 2, Grade B) [20]. Collection of serum samples for TT determinations is preferred between the morning hours of 7:00 and 11:00 [21] (evidence level: Level 2a, Grade A). Free testosterone or BT should be considered when the TT concentration does not correspond with clinical presentation, since individual variation in sex hormone binding globulin (SHBG) concentrations may influence TT results. Measurement of serum SHBG, together with a reliable measurement of TT, allows for the determination of the calculated free T level (evidence level: Level 2b, Grade A) [22].

### 2.2. EAU Guidelines

Low levels of circulating androgens may be associated with specific signs and symptoms. The most prevalent symptoms of LOH are reduced sexual desire and sexual activity, ED, loss of vigor, and mood changes [23]. Signs and symptoms of LOH vary depending on age of onset, duration, and severity of the deficiency. Reference ranges for the lower normal level of T (2.5%) have been compiled from three large community-based samples and suggest a cut-off of 12.1 nmol/L for TT and 243 pmol/L for FT, to distinguish between normal levels and levels possibly associated with deficiency [24]. It should, however, be noted that these symptoms are also found in men with normal T levels and may have causes other than androgen deficiency. In men aged 40–79 years, the strongest predictor for hypogonadism was three sexual symptoms (decreased sexual thoughts, weakened morning erections, and ED) and either a TT level of <8 nmol/L, or serum T in the range of 8–11 nmol/L, and FT <220 pmol/L [25]. 

Laboratory testing of T should reflect the diurnal variation of testosterone. In most cases, two morning (7:00–11:00) samples are sufficient, but further evaluation should be triggered if the difference between the two measurements is >20 percent [26]. In cases with discrepancy between T levels and symptoms, the FT level should be analyzed [15]. For determination of FT levels, the calculation of FT, with the help of SHBG, is recommended [15].

In LOH, published questionnaires are unreliable, have low specificity, and are not effective for case findings [27]. Assessment of BMI, the waist:hip ratio, body hair, male pattern hair loss, presence of gynecomastia, testicular size, and examination of the penis, as well as examination of prostate should be included [15].

### 2.3. ESE and EAA Guidelines

The ESE and the EAA suggest against routinely prescribing TRT for all men 65 years or older with low T concentrations [18]. TRT should be recommended in men >65 years who have symptoms or conditions suggestive of T deficiency (such as low libido or unexplained anemia) and consistently and unequivocally low morning T. Low libido, ED, and less specific symptoms (such as fatigue, irritability, depressed mood, poor concentration, reduced physical performance, and sleep disturbance) are associated with low T concentrations [28]. In the European Male Aging Study (a cohort study of community-dwelling middle-aged and older men in Europe), only sexual symptoms (poor morning erections, decreased libido, and ED) had a syndromic association with TT concentrations of 320 ng/dL (11 nmol/L) and FT of 64 pg/mL (220 pmol/L) (after adjusting for age) [7]. Testosterone alone is required to maintain lean mass and muscle size and strength; estradiol is required to prevent increases in fat mass and vasomotor symptoms; and both T and estradiol are required to maintain sexual function and bone mineral density (BMD) [29]. Testosterone concentrations exhibit significant diurnal and day-to-day variations and may be suppressed by food intake or glucose. Therefore, clinicians should measure TT concentrations on two separate mornings when the patient is fasting. Clinicians should use an accurate and reliable method, optimally an assay that has been certified by an accuracy-based standardization.

### 2.4. AUA Guidelines

Clinicians should use a TT level below 300 ng/dL as a reasonable cut-off in support of the diagnosis of low testosterone (moderate recommendation; evidence level: Grade B) [30]. The diagnosis of low T should be made only after two TT measurements are taken on separate occasions, with both conducted in early morning (strong recommendation; evidence level: Grade A). The clinical diagnosis of T deficiency is only made when patients have low TT levels combined with symptoms and/or signs (moderate recommendation; evidence level: Grade B). Clinicians should consider measuring TT in patients with a history of unexplained anemia, bone density loss, diabetes, exposure to chemotherapy, exposure to testicular radiation, HIV/AIDS, chronic narcotic use, male infertility, pituitary dysfunction, and chronic corticosteroid use, even in the absence of signs or symptoms associated with T deficiency (moderate recommendation; evidence level: Grade B). The use of validated questionnaires is not currently recommended to either define which patients are candidates for T therapy or to monitor symptom response in patients on T therapy (conditional recommendation; evidence level: Grade C). In the patients with low testosterone, clinicians should measure serum luteinizing hormone levels (strong recommendation; evidence level: Grade A). Serum prolactin levels should be measured in patients with low T levels, combined with low or low/normal luteinizing hormone levels (strong recommendation; evidence level: Grade A). Patients with persistently high prolactin levels of unknown etiology should undergo evaluation for endocrine disorders (strong recommendation; evidence level: Grade A). Serum estradiol should be measured in T-deficient patients who present with breast symptoms or gynecomastia, prior to the commencement of T therapy (expert opinion). Men with T deficiency who are interested in fertility should have a reproductive health evaluation performed prior to treatment (moderate recommendation; evidence level: Grade B). Prostate-specific antigen (PSA) should be measured in men over 40 years of age prior to commencement of T therapy to exclude a prostate cancer diagnosis (clinical principle).

### 2.5. Canadian Men’s Health Foundation Multidisciplinary Guidelines

Diagnosis of T deficiency syndrome requires the presence of the clinical manifestations of T deficiency, together with documented T levels below the local laboratory ranges [31]. The initial biochemical test should be TT level measured in serum samples taken in the morning; determinations of BT or FT should be restricted to patients with equivocally low TT levels (strong recommendation; high-quality evidence). T levels should be measured with the use of T assays traceable to internationally-recognized standardized reference material; commercial assays should be certified by the testosterone standardization program of the U.S. Centers for Disease Control and Prevention (strong recommendation; high-quality evidence). In men with ED and no other manifestations of T deficiency syndrome, they suggest investigation. Treatment is recommended for T deficiency syndrome; the choice of treatment is based on product safety, efficacy, tolerability, cost, and the absence of contraindications. Men with testosterone deficiency syndrome and stable cardiovascular disease (CVD) are candidates for TRT (weak recommendation; low-quality evidence). Hypogonadal men with successfully-treated prostate cancer (PCa) may be candidates for T supplementation; these patients require referral to a specialist, because treatment involves close monitoring by a physician with expertise in the risks and benefits of T therapy. Men with a history of breast cancer are not candidates for testosterone replacement therapy (weak recommendation; moderate-level of evidence). They recommend treatment with a PDE-5 inhibitor in men with T deficiency syndrome and persistent ED that is adequately treated with T (strong recommendation; high-quality evidence). Regular monitoring for clinical and biochemical response, and for adverse effects, to TRT is essential, particularly during the first year of treatment.

## 3. Optimal Indication for TRT

TRT for LOH is increasing every year; however, it is clear from the literature that many men are using T without a clear indication [32,33,34]. Previous studies estimated that up to 25% of men who received TRT did not have their T tested prior to initiation of TRT [32,33]. Given the growing concern and need for proper TRT, a need exists to define a TT threshold to guide clinicians in the diagnosis and management of LOH patients. Based on the recent guidelines by ISSAM, EAU, ESE, EAA, and AUA, a TT of 250–350 ng/dL is the proper threshold value to define low testosterone [13,15,18,30] (Table 2). Adherence to this recommendation can increase clinicians’ confidence regarding the risk:benefit ratio of TRT, explicitly placing a higher value on maximizing true benefit and reducing clinically-inappropriate use of TRT (Figure 1). The prevalent confusion about the diagnosis of T deficiency, the inappropriate use of TRT, and the recent reports of potentially increased risk of serious adverse effects with the use of TRT indicate an important need for guidance among health professionals in the management of testosterone deficiency. Although direct measurement of FT has a generally good correlation with equilibrium dialysis, it is not reliable because of a high coefficient of variation [12,35]. Given that the direct method for FT measurement is also time consuming and labor intensive, recent guidelines do not recommend using FT measurements as the primary diagnostic method for LOH [13,15,18,30]. However, in cases with discrepancy between T levels and symptoms, FT levels should be analyzed [15].

There are inherent challenges to T measurements due to the health status of patients at the time of testing, circadian rhythms in T production, intra-individual variability, and inconsistencies in the assays themselves. To ensure accuracy and precision, it is necessary to obtain at least two serum TT measurements in early morning to diagnose patients with LOH [13,15,18,30]. The sample collection for T measurement occurs between 7:00 and 11:00, or within 3 h after waking in the case of shift workers [31]. Intra-individual T variability is significant, and repeated measures can fluctuate 65–153% between tests, depending on the assay used; however, performing two or three measurements can reduce this variability by 30–43%, respectively [36]. T levels should be measured with the use of T assays traceable to internationally-recognized standardized reference materials [31].

Low TT alone does not define LOH. The diagnosis of LOH must include the presence of signs and/or symptoms associated with low T, in combination with documented low TT levels (Table 1). Clinicians should refrain from measuring T levels in patients who are asymptomatic, do not exhibit signs related to low T, or do not have any comorbid conditions that are associated with low T.

Several validated questionnaires are used as screening tools to identify men of LOH, but there is an absence of concordance among the questionnaires as to which symptoms are related to low T or to what extent these symptoms improve with TRT. Specificities and sensitivities vary greatly among these tests, making them ill-suited for screening or for use as a surrogate for T laboratory testing. Higher sensitivities and lower specificities have been reported for the AMS and ADAM, with sensitivities/specificities of 81%/19% and 97%/39%, respectively, for each questionnaire; while the MMAS and ANDROTEST exhibit lower sensitivities and higher specificities, with sensitivities/specificities of 60%/53% and 71%/65%, respectively [37].

Prior to commencing TRT, all patients should undergo a baseline measurement of hemoglobin (Hb)/hematocrit (Hct). If the Hct exceeds 50%, clinicians should consider withholding TRT until the etiology of the high Hct is explained [38]. Men with elevated Hct and on-treatment low/normal total and free T levels should be referred to a hematologist for further evaluation, and possible coordination of phlebotomy. Testosterone has a stimulating effect on erythropoiesis, and elevation of Hb/Hct is the most frequent adverse event related to TRT [39]. During TRT, levels of Hb/Hct generally rise for the first six months, then tend to plateau [40].

PSA should be measured in men over 40 years of age prior to commencement of TRT to exclude a prostate cancer diagnosis [41]. PSA secretion is an androgen-dependent phenomenon, and the rise of PSA levels in patients on TRT is primarily dependent on baseline TT levels. A previous study found that men with lower baseline T levels were more likely to experience PSA level increases after TRT [42]. In that study, participants (*N* = 451) received 5–10 g of 1% T gel daily for 12 months. Patients were divided into two groups: A (*n* = 197 with TT <250 ng/dL) and B (*n* = 254 with TT ≥250 ng/dL). In Group A, but not Group B, baseline PSA levels correlated significantly with total testosterone levels (*r* = 0.2; *p* < 0.01). At the end of follow-up, PSA increased significantly in Group A (21.9% change; 0.19 ± 0.61 ng/mL; *p* = 0.02), but not in Group B (14.1% change; 0.28 ± 1.18 ng/mL; *p* = 0.06), with the greatest PSA change observed after one month of treatment [42].

The effects of TRT on cardio vascular disease (CVD) remain a point of concern. Ferrucci et al. showed that low T levels had an independent influence on the development of anemia in older adults [43]. T stimulates the production of erythropoietin-responsive cells and burst-forming units in the bone marrow, which boosts iron absorption and erythropoiesis [44]. The effects of T on the bone marrow affecting the hematopoietic growth factors and iron absorption show the association between T and erythropoiesis. In an earlier study, our data demonstrated that subjects with low TT and FT levels had low Hb and Hct levels [45]. This result suggests that TT and FT may play a significant role in erythropoiesis. According to a recent study, TRT in older men with low T levels significantly increased Hb levels of those with unexplained anemia, as well as those with anemia from known causes [46]. Measurement of T levels might be considered in men 65 years or older who have unexplained anemia and symptoms of low T levels. Although men with hypogonadism were not always anemic in our previous study, the association between low testosterone and low Hb levels was statistically significant [45]. Furthermore, our previous study noted that the prevalence of anemia decreased, and patients with anemia showed increased erythropoietin after TRT [47]. TRT may be effective in men with hypogonadism to reduce the incidence of anemia and the CVD associated with anemia [47,48].

Little is yet known about the clinical utility of serum T levels as a predictor of disease progression in PCa patients. Recently, Ferro et al. [49] showed that low serum T levels (<300 ng/dL) were significantly associated with upgrading, upstaging, unfavorable disease, and positive surgical margins in PCa patients. On this basis, TT should be measured in patients with a localized PCa, in particularly when active surveillance or nerve-sparing surgery is considered [49]. Furthermore, growing evidence supports the idea that a decreased serum T concentration, related to different metabolic disorders including obesity and metabolic syndrome, may modulate PCa aggressiveness [50]. Furthermore, Cobeli et al. [51] showed that BMI was significantly associated with upgrading, upstaging, and seminal vesicle invasion in men with low-risk PCa. These data further support the idea that obesity is associated with PCa aggressiveness [50,51].

## 4. Conclusions

The diagnosis of LOH requires the presence of characteristic signs and symptoms, in combination with decreased serum TT. Based on the recent guidelines by ISSAM, EAU, ESE, EAA, and AUA, a TT of 250–350 ng/dL is the proper threshold value to define low testosterone. The optimal indication of TRT for LOH is the presence of signs and symptoms and of hypogonadism and low T without contraindication for TRT.

## Figures and Tables

**Figure 1 jcm-08-00209-f001:**
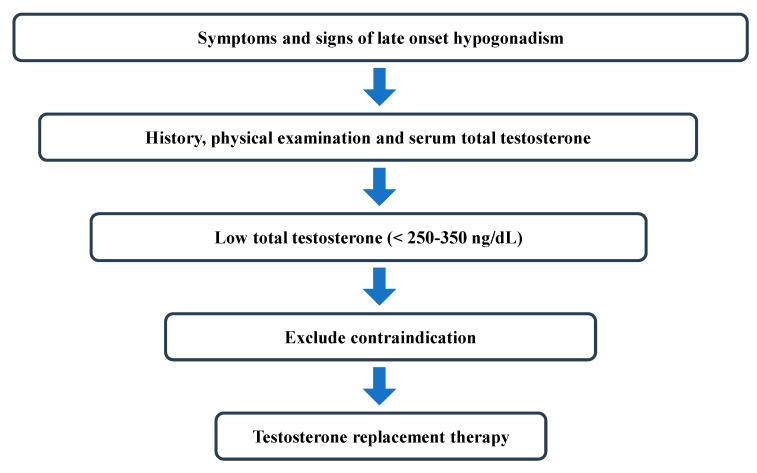
Flowchart for the management of late-onset hypogonadism.

**Table 1 jcm-08-00209-t001:** Symptoms and signs associated with LOH.

Physical	Cognitive	Sexual
Anemia Reduced energy	Depressive symptoms	Reduced sex drive
Reduced endurance	Cognitive dysfunction	Reduced erectile function
Diminished work performance	Reduced motivation	
Diminished physical performance	Poor concentration	
Loss of body hair	Poor memory	
Reduced beard growth	Irritability	
Fatigue		
Reduced lean muscle mass		
Obesity		

**Table 2 jcm-08-00209-t002:** The guidelines for the diagnosis of the optimal indication of treatment for LOH.

Guideline	Threshold of TT	Threshold of FT	How Many Times T Needs to Measured	Questionnaire
ISSAM	12.1 nmol/L	225 pmol/L	Not suggested	Recommended
EAU	12.1 nmol/L	243 pmol/L	2 times	Not recommended
ESE and EAA	320 ng/dL	220 pmol/L	2 times	Not recommended
AUA	300 ng/dL	Not suggested	2 times	Not recommended
Canadian Men’s Health Foundation	Not suggested	Not suggested	Not suggested	Not recommended

LOH: late-onset hypogonadism, T: testosterone, TT: total testosterone, FT: free testosterone, ISSAM: International Society for the Study of Aging Male, EAU: European Association of Urology, ESE: European Society of Endocrinology, EAA: European Academy of Andrology, AUA: American Association of Urology.

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
