# Peer review of "The Optimal Indication for Testosterone Replacement Therapy in Late Onset Hypogonadism"

_jcm, 2019, doi:10.3390/jcm8020209_

Reviewer 1 Report

This is an interisting review refocus on omtimal managment of T replacement in LOH , I suggest to extend the field of your research also to oncological aspects in PCa risk and to add a critical review on obesity impact in T modulation and Pc cancer .I suggest to add in discussion these aspects as define by :

Body mass index was associated with upstaging and upgrading in patients with low-risk prostate cancer who met the inclusion criteria for active surveillance.

de Cobelli O Ur oncology 2015

Low serum total testosterone level as a predictor of upstaging and upgrading in low-risk prostate cancer patients meeting the inclusion criteria for active surveillance.

Ferro M et al -2017

Author Response

January 24, 2019.

Dear, reviewer of Journal of Clinical Medicine

We are pleased to submit our paper entitled by “The Optimal Indication for Testosterone Replacement Therapy in Late Onset Hypogonadism” for consideration for publication in Investigative and Clinical Urology. This version has been revised on accordance with recommendation from you and the referee whose comments were sent me on January 23, 2019. I considered the referee suggestions for great value and appreciated the kind attention given to this paper. I hope that you and the referees will now find the manuscript satisfactory with the following changes.

To help you and referees to evaluate the changes I enclosed the followings:

1. Revised version of marked manuscript.

In marked manuscript, we used words of red color for revision.

2. Reply sheets.

I wish to thank you again for your endless patience with this paper.

PEER-REVIEW COMMENTS:

This is an interisting review refocus on optimal managment of T replacement in LOH , I suggest to extend the field of your research also to oncological aspects in PCa risk and to add a critical review on obesity impact in T modulation and Pc cancer .I suggest to add in discussion these aspects as define by :.
Body mass index was associated with upstaging and upgrading in patients with low-risk prostate cancer who met the inclusion criteria for active surveillance.

de Cobelli O Ur oncology 2015

Low serum total testosterone level as a predictor of upstaging and upgrading in low-risk prostate cancer patients meeting the inclusion criteria for active surveillance.

Ferro M et al -2017

Respond to comment

Thank you for your detailed review and kind consideration. We add critical review on obesity impact in T modulation and Pc cancer as your comments in discussion.

We added sentence in our manuscript as below.

Little is yet known about the clinical utility of serum T levels as a predictor of disease progression in PCa patients. Recently, Ferro et al [49] showed that low serum T levels (<300 ng/dL) were significantly associated with upgrading, upstaging, unfavorable disease and positive surgical margins in PCa patients. On this basis, TT should be measured in patients with a localized PCa, in particularly when AS or nerve-sparing surgery is considered [49]. Furthermore, growing evidence supports the idea that a decreased serum T concentration, related to different metabolic disorders including obesity and metabolic syndrome, may modulate PCa aggressiveness [50]. Also, Cobeli et al [51] showed that BMI was significantly associated with upgrading, upstaging, and seminal vesicle invasion in men with low-risk PCa. These data further support the idea that obesity is associated with PCa aggressiveness [50,51].

References

Ferro, M,; Lucarelli, G,; Bruzzese, D,; Di Lorenzo, G,; Perdonà, S,; Autorino, R,; Cantiello, F,; La Rocca, R,; Busetto, GM,; Cimmino, A,; et al. Low serum total testosterone level as a predictor of upstaging and upgrading in low-risk prostate cancer patients meeting the inclusion criteria for active surveillance. Oncotarget. 2017, 8, 18424–18434.

Bhindi, B,; Locke, J,; Alibhai, SM,; Kulkarni, GS,; Margel, DS,; Hamilton, RJ,; Finelli, A,; Trachtenberg, J,; Zlotta, AR,; Toi, A,; et al. Dissecting the association between metabolic syndrome and prostate cancer risk: analysis of a large clinical cohort. Eur Urol. 2015, 67, 64-70.

de Cobelli, O,; Terracciano, D,; Tagliabue, E,; Raimondi, S,; Galasso, G,; Cioffi, A,; Cordima, G,; Musi, G,; Damiano, R,; Cantiello, F,; et al. Body mass index was associated with upstaging and upgrading in patients with low-risk prostate cancer who met the inclusion criteria for active surveillance. Urol Oncol. 2015, 33,201.e1–201.e8.

Reviewer 2 Report

I think the authors should discuss the cardiovascular risks of TRT. Several studies have demonstrated that TRT to men with CVD increases risk of CV events considerably

Author Response

January 24, 2019.

Dear, reviewer of Journal of Clinical Medicine

We are pleased to submit our paper entitled by “The Optimal Indication for Testosterone Replacement Therapy in Late Onset Hypogonadism” for consideration for publication in Investigative and Clinical Urology. This version has been revised on accordance with recommendation from you and the referee whose comments were sent me on January 23, 2019. I considered the referee suggestions for great value and appreciated the kind attention given to this paper. I hope that you and the referees will now find the manuscript satisfactory with the following changes.

To help you and referees to evaluate the changes I enclosed the followings:

1. Revised version of marked manuscript.

In marked manuscript, we used words of red color for revision.

2. Reply sheets.

I wish to thank you again for your endless patience with this paper.

PEER-REVIEW COMMENTS:

I think the authors should discuss the cardiovascular risks of TRT. Several studies have demonstrated that TRT to men with CVD increases risk of CV events considerably

Respond to comment

Thank you for your detailed review and kind consideration. However, the effects of TRT on cardiovascular health remain a point of concern. Although participants with low testosterone levels were not always anemic, the association between low testosterone and decreased hemoglobin was statistically significant. These results suggested that low testosterone is one of the causal factors of anemia. Testosterone has an important role in erythropoiesis as it stimulates the production of erythropoietin responsive cells and burst forming units in bone marrow. In our opinion, low testosterone may be a cause of anemia, and our results suggest that TRT in hypogonadal patients may be effective in reducing the anemia and the CVD associated with anemia.

References

Ferrucci, L,; Maggio, M,; Bandinelli, S,; Basaria, S,; Lauretani, F,; Ble, A,; Valenti, G,; Ershler, WB,; Guralnik, JM,; Longo, DL. Low testosterone levels and the risk of anemia in older men and women. Arch Intern Med. 2006, 166, 1380–1388.

Shahani, S,; Braga-Basaria, M,; Maggio, M,; Basaria, S. Androgens and erythropoiesis: past and present. J Endocrinol Invest. 2009, 32, 704–716.

Shin, YS,; You, JH,; Cha, JS,; Park, JK. The relationship between serum total testosterone and free testosterone levels with serum hemoglobin and hematocrit levels: a study in 1221 men. Aging Male. 2016, 19, 209-214.

Roy, CN,; Snyder, PJ,; Stephens-Shields, AJ,; Artz, AS,; Bhasin, S,; Cohen, HJ,; Farrar, JT,; Gill, TM,; Zeldow, B,; Cella, D,; et al. Association of Testosterone Levels With Anemia in Older Men: A Controlled Clinical Trial. JAMA Intern Med. 2017, 177, 480-490.

5. Zhang, LT,; Shin, YS,; Kim, JY,; Park, JK. Could testosterone replacement therapy in hypogonadal men ameliorate anemia, a cardiovascular risk factor? An observational, 54-week cumulative registry study. J Urol. 2016, 195, 1057–1064.

We added sentence in our manuscript as below.

The effects of TRT on cardio vascular disease (CAD) remain a point of concern. Ferrucci et al. showed that low T levels had an independent influence on the development of anemia in older adults [43]. T stimulates the production of erythropoietin-responsive cells and burst-forming units in the bone marrow, which boosts iron absorption and erythropoiesis [44]. The effects of T on the bone marrow affecting the hematopoietic growth factors and iron absorption, show the association between T and erythropoiesis. In an earlier study, our data demonstrated that subjects with low TT and FT levels had low Hb and Hct levels [45]. This result suggests that TT and FT may play a significant role in erythropoiesis. According to a recent study, TRT in older men with low T levels, significantly increased Hb levels of those with unexplained anemia, as well as those with anemia from known causes [46]. Measurement of T levels might be considered in men 65 years or older who have unexplained anemia and symptoms of low T levels. Although men with hypogonadism were not always anemic in our previous study, the association between low testosterone and low Hb levels was statistically significant [45]. Also, our previous study noted that the prevalence of anemia decreased, and patients with anemia showed increased erythropoietin after TRT [47]. TRT may be effective in men with hypogonadism to reduce the incidence of anemia and the CVD associated with anemia [47,48].

Reviewer 3 Report

The authors present a well-written summary of different societies’ guidelines for diagnosis and management of low testosterone.  Overall, it is well written but slightly repetitive and could be trimmed to be more efficient in conveying the guidelines.  For example, evidence regarding questionnaires does not need to be repeated under each different society’s guidelines.  Below are my specific comments.

-How did the authors choose which guidelines to study/summarize?  Why were American Endocrine and Canadian guidelines not included?  I ask because the Canadian Medical Association Journal has published guidelines regarding testosterone, and they do not specify a number to use as a cutoff for low testosterone (http://wwwNaNaj.ca/content/187/18/1369).  These guidelines warrant inclusion in this manuscript, as it is unique.

-The first line of the introduction states that symptoms of low T are “mostly of a sexual nature.”  This is inaccurate, as is indicated by the more detailed description of signs and symptoms of low T later in the manuscript, and should be modified.

-The authors state, “Demographic data clearly demonstrate that the percentage of older adults in the general population is increasing.”  Specific numbers should be cited here.

-On page 4, in the last paragraph under “3. Optimal indication for TRT,” a couple of sentences have grammar/syntax errors.  Please proofread.

-The authors state that following these thresholds would reduce clinically inefficacious TRT.  Please expand and provide evidence for “inefficacious TRT.”

-More discussion should be given to different laboratory testing modalities of testosterone.

-Anemia should be included in Table 1.

-Table 2 should be expanded to discuss what other stipulations for diagnosis, such as how often T should be measured, etc.

Author Response

January 24, 2019.

Dear, reviewer of Journal of Clinical Medicine

We are pleased to submit our paper entitled by “The Optimal Indication for Testosterone Replacement Therapy in Late Onset Hypogonadism” for consideration for publication in Investigative and Clinical Urology. This version has been revised on accordance with recommendation from you and the referee whose comments were sent me on January 23, 2019. I considered the referee suggestions for great value and appreciated the kind attention given to this paper. I hope that you and the referees will now find the manuscript satisfactory with the following changes.

To help you and referees to evaluate the changes I enclosed the followings:

1. Revised version of marked manuscript.

In marked manuscript, we used words of red color for revision.

2. Reply sheets.

I wish to thank you again for your endless patience with this paper.

PEER-REVIEW COMMENTS 1:

The authors present a well-written summary of different societies’ guidelines for diagnosis and management of low testosterone.  Overall, it is well written but slightly repetitive and could be trimmed to be more efficient in conveying the guidelines.  For example, evidence regarding questionnaires does not need to be repeated under each different society’s guidelines.  Below are my specific comments.

-How did the authors choose which guidelines to study/summarize?  Why were American Endocrine and Canadian guidelines not included?  I ask because the Canadian Medical Association Journal has published guidelines regarding testosterone, and they do not specify a number to use as a cutoff for low testosterone (http://wwwNaNaj.ca/content/187/18/1369).  These guidelines warrant inclusion in this manuscript, as it is unique.
Respond to comment 1

Thank you for your detailed review and kind consideration. We add Canadian Men’s Health Foundation Multidisciplinary Guidelines in our study.

We added sentence in our manuscript as below.

(5) Canadian Men’s Health Foundation Multidisciplinary guidelines [30]

Diagnosis of T deficiency syndrome requires the presence of the clinical manifestations of T deficiency, together with documented T levels below the local laboratory ranges. The initial biochemical test should be TT level measured in serum samples taken in the morning; determinations of BT or FT should be restricted to patients with equivocally low TT levels (strong recommendation; high-quality evidence). T levels should be measured with the use of T assays traceable to internationally recognized standardized reference material; commercial assays should be certified by the testosterone standardization program of the US Centers for Disease Control and Prevention (strong recommendation; high-quality evidence). In men with erectile dysfunction and no other manifestations of T deficiency syndrome, they suggest investigation. Treatment is recommended for T deficiency syndrome; the choice of treatment is based on product safety, efficacy, tolerability, cost and the absence of contraindications. TRT is appropriate in men with T deficiency syndrome who have CAD or are at risk of CAD. Hypogonadal men with successfully treated prostate cancer (PCa) may be candidates for T supplementation; these patients require referral to a specialist, because treatment involves close monitoring by a physician with expertise in the risks and benefits of T therapy. Men with a history of breast cancer are not candidates for testosterone replacement therapy (weak recommendation; moderate-level of evidence). They recommend treatment with a PDE-5 inhibitor in men with testosterone deficiency syndrome and persistent erectile dysfunction that is adequately treated with testosterone (strong recommendation; high-quality evidence). Regular monitoring for clinical and biochemical response, and for adverse effects, to TRT is essential, particularly during the first year of treatment.

References

Morales, A,; Bebb, RA,; Manjoo, P,; Assimakopoulos, P,; Axler, J,; Collier, C,; Elliott, S,; Goldenberg, L,; Gottesman, I,; Grober, ED,; et al. Diagnosis and management of testosterone deficiency syndrome in men: clinical practice guideline. CMAJ. 2015, 187, 1369-1377.

PEER-REVIEW COMMENTS 2:

-The first line of the introduction states that symptoms of low T are “mostly of a sexual nature.”  This is inaccurate, as is indicated by the more detailed description of signs and symptoms of low T later in the manuscript, and should be modified.

Respond to comment 2

Thank you for your detailed review and kind consideration. We revise our introduction as your comments.

We added sentence in our manuscript as below.

Late-onset hypogonadism (LOH) is characterized by low testosterone (T) levels and clinical symptoms [1-6]. Sexual symptoms and fatigue are the earliest and most common presentations [1-3]. Other symptoms include depression, sleep alterations, poor concentration and metabolic disorders are seen at borderline T levels [4-6].

PEER-REVIEW COMMENTS 3:

-The authors state, “Demographic data clearly demonstrate that the percentage of older adults in the general population is increasing.”  Specific numbers should be cited here.

Respond to comment 3

Thank you for your detailed review and kind consideration. We add reference.

Waite, LJ. The Demographic Faces of the Elderly. Popul Dev Rev. 2004;30:3-16.

PEER-REVIEW COMMENTS 4:

-On page 4, in the last paragraph under “3. Optimal indication for TRT,” a couple of sentences have grammar/syntax errors.  Please proofread.

Respond to comment 4

Thank you for your detailed review and kind consideration.

The English editing was done by English editing service.

PEER-REVIEW COMMENTS 5:

-The authors state that following these thresholds would reduce clinically inefficacious TRT.  Please expand and provide evidence for “inefficacious TRT.”

Respond to comment 5

Thank you for your detailed review and kind consideration. We revise our manuscript as your comments.

We added sentence in our manuscript as below.

Adherence to this recommendation can increase clinicians' confidence regarding the risk-benefit ratio of TRT, explicitly placing a higher value on maximizing true benefit, and reducing clinically inappropriate use of TRT (Figure 1). The prevalent confusion about the diagnosis of T deficiency, the inappropriate use of TRT and the recent reports of potentially increased risk of serious adverse effects with the use of TRT indicate an important need for guidance among health professionals in the management of testosterone deficiency.

PEER-REVIEW COMMENTS 6:

More discussion should be given to different laboratory testing modalities of testosterone.

Respond to comment 6

Thank you for your detailed review and kind consideration. We revise our manuscript as your comments.

We added sentence in our manuscript as below.

There are inherent challenges to T measurements due to the health status of patients at the time of testing, circadian rhythms in T production, intra-individual variability, and inconsistencies in the assays themselves. To ensure accuracy and precision, it is necessary to obtain at least two serum TT measurements in early morning to diagnose patients with LOH [13,15,18,30]. The sample collection for T measurement occur between 7 am and 11 am, or within 3 hours after waking in the case of shift workers [31]. Intra-individual T variability is significant and repeated measures can fluctuate 65 to 153% between tests, depending upon the assay used, however performing two or three measurements can reduce this variability by 30 to 43%, respectively [36]. T levels should be measured with the use of T assays traceable to internationally recognized standardized reference materials [31].

PEER-REVIEW COMMENTS 7:

Anemia should be included in Table 1.

Respond to comment 7

Thank you for your detailed review and kind consideration. We add anemia in Table 1.

PEER-REVIEW COMMENTS 8:

Table 2 should be expanded to discuss what other stipulations for diagnosis, such as how often T should be measured, etc.

Respond to comment 8

Thank you for your detailed review and kind consideration. However, Table 2 is list of

optimal indication of treatment for LOH. How often T should be measured should be include in TRT monitoring.

Round  2

Reviewer 3 Report

The authors have done a nice job addressing most of my comments in a clear, well-organized fashion.

-“Table 2 should be expanded to discuss what other stipulations for diagnosis, such as how often T should be measured, etc.”  My main point was not to discuss monitoring during therapy, but rather how many times testosterone needs to be measured to make a diagnosis, per each guideline.  For example, some guidelines explicitly state 2 measurements are needed, while some do not.  This would be an interesting column to add to Table 2 and would make it more widely citable.  However, I will defer to the authors regarding this.

Author Response

February 1, 2019.

Dear, reviewer

We are pleased to submit our paper entitled by “The Optimal Indication for Testosterone Replacement Therapy in Late Onset Hypogonadism” for consideration for publication in Investigative and Clinical Urology. This version has been revised on accordance with recommendation from you and the referee whose comments were sent me on January 31, 2019. I considered the referee suggestions for great value and appreciated the kind attention given to this paper. I hope that you and the referees will now find the manuscript satisfactory with the following changes.

To help you and referees to evaluate the changes I enclosed the followings:

Revised version of marked manuscript.

In marked manuscript, we used words of red color for revision.

Reply sheets.

I wish to thank you again for your endless patience with this paper.

Very Respectfully,

Jong Kwan Park, M.D., Ph.D.

Department of Urology, Chonbuk National University Medical School, 560-180, Jeonju, South Korea

Tel: 82-63-245-1510

Fax: 82-63-250-1564

E-mail:[email protected]

PEER-REVIEW COMMENTS:

The authors have done a nice job addressing most of my comments in a clear, well-organized fashion.

-“Table 2 should be expanded to discuss what other stipulations for diagnosis, such as how often T should be measured, etc.”  My main point was not to discuss monitoring during therapy, but rather how many times testosterone needs to be measured to make a diagnosis, per each guideline.  For example, some guidelines explicitly state 2 measurements are needed, while some do not.  This would be an interesting column to add to Table 2 and would make it more widely citable.  However, I will defer to the authors regarding this.
Respond to comment

Thank you for your detailed review and kind consideration. We add ‘how often T should be measured’ in the Table. 2 as below

Table 2. The guidelines for diagnosis of optimal indication of treatment for LOH.

Guideline

Threshold of TT

Threshold of FT

How many   times T needs to measured

Questionnaire

ISSAM

12.1 nmol/L

225 pmol/L

Not suggested

Recommended

EAU

12.1   nmol/L

243   pmol/L

2 times

Not recommended

ESE and EAA

320 ng/dL

220 pmol/L

2 times

Not recommended

AUA

300   ng/dL

Not suggested

2 times

Not recommended

Canadian   Men’s Health Foundation

Not suggested

Not suggested

Not suggested

Not recommended

T: Testosterone, TT: Total testosterone, FT: Free testosterone, ISSAM: International Society for the Study of Aging Male, EAU: European Association of Urology, ESE: European Society of Endocrinology, EAA: European Academy of Andrology, AUA: American Association of Urology.